# Evaluation of Electrospun Self-Supporting Paper-Like Fibrous Membranes as Oil Sorbents

**DOI:** 10.3390/membranes11070515

**Published:** 2021-07-08

**Authors:** Adele Folino, Claudia Triolo, Beatrix Petrovičová, Fabiolo Pantò, Demetrio A. Zema, Saveria Santangelo

**Affiliations:** 1Department of Agriculture, Mediterranean University of Reggio Calabria, Località Feo di Vito, I-89122 Reggio Calabria, Italy; adelefolino@alice.it; 2Department of Civil, Energy, Environmental and Materials Engineering (DICEAM), Mediterranean University, Località Feo di Vito, I-89122 Reggio Calabria, Italy; claudia.triolo@unirc.it (C.T.); beatrix.petrovicova@unirc.it (B.P.); 3Institute of Advanced Technologies for Energy (ITAE), Italian National Research Council (CNR), I-98126 Messina, Italy; fabiola.panto@itae.cnr.it

**Keywords:** electrospinning, oil spill, recovery efficiency, oil/water selectivity

## Abstract

Presently, adsorption/absorption is one of the most efficient and cost-effective methods to clean oil spill up. In this work, self-supporting paper-like fibrous membranes were prepared via electrospinning and carbonisation at different temperatures (500, 650 or 800 °C) by using polyacrylonitrile/polymethylmethacrylate blends with a different mass ratio of the two polymers (1:0, 6:1 or 2:1). After morphological and microstructural characterisation, the as-produced membranes were evaluated as sorbents by immersion in vegetable (sunflower seed or olive) and mineral (motor) oil or in 1:4 (*v*:*v*) oil/water mixture. Nitrogen-rich membrane carbonised at the lowest temperature behaves differently from the others, whose sorption capacity by immersion in oil, despite the great number of sorbent and oil properties involved, is mainly controlled by the fraction of micropores. The encapsulation of water nanodroplets by the oil occurring during the immersion in oil/water mixture causes the oil-from-water separation ability to show an opposite behaviour compared to the sorption capacity. Overall, among the investigated membranes, the support produced with 2:1 mass ratio of the polymers and carbonisation at 650 °C exhibits the best performance both in terms of sorption capacity (73.5, 54.8 and 12.5 g g^−1^ for olive, sunflower seed and motor oil, respectively) and oil-from-water separation ability (74, 69 and 16 for olive, sunflower seed and motor oil, respectively).

## 1. Introduction

Contamination by mineral and vegetable oils represents a serious environmental concern and a critical health hazard for ecosystems’ life in soils, freshwater and seawater [1,2,3]. Organic compounds, such as polyphenols, hinder the treatment of water streams in the traditional depuration plants, due to the resistance of these molecules to biodegradation [4]. Clean-up treatments for oil spills are based on mechanical, chemical and biological methods [5,6,7,8,9,10,11,12]. Most of them are complex and expensive and show a low depuration efficiency [9,12,13]. Adsorption/absorption seems to be the most effective and low-cost process for the spill remediation purpose [10]. Natural [14] and synthetic polymer-based [6,15,16,17,18,19,20,21,22,23,24] fibrous membranes show good water purification and oil sorption capacity, buoyancy and resistance to rot and mildew [25,26]. Chemically inert polymer-based fibrous membranes with large surface areas and hierarchical porosity, which are beneficial for liquid sorption [27,28], can be easily produced via electrospinning, a cost-effective and scalable top-down technique [10,19,29,30,31,32,33,34,35,36,37,38,39]. Compared to the low-cost melt-blown membranes [40], the electrospun supports show greater re-usability in filtration applications. Several polymers, such as polystyrene [10], polyimide [27] and polyvinylpyrrolidone [41], have been evaluated for the production of sorbent membranes.

Presently, due to new environmental protection policies, the development of materials with high and efficient sorption capacity is attracting renewed attention [42,43,44]. Although polyacrylonitrile (PAN) is among the polymers most frequently chosen for the production of electrospun fibrous membranes [31], few studies have reported the use of PAN-derived membranes for oil recovery (Appendix A), focusing mainly on mineral oils [5,29,45]. Usually, to increase the surface area and porosity of the sorbent and improve its sorption performance, after synthesis, the membranes are treated with harsh chemicals [46,47,48], which are dangerous for human health and the conservation of ecosystems.

This paper deals with the preparation and evaluation as oil sorbents of self-supporting paper-like PAN-derived fibrous membranes. To limit the environmental impact, prevent heavy harmful effects on human health and improve the sustainability of the sorbent production process, the increase in the surface area of sorbent and porosity is achieved by incorporating a porogen (polymethylmethacrylate, PMMA) in the spinnable solution [46,47,48] and the as-carbonised membranes are evaluated as sorbents of both mineral (motor) and vegetable (sunflower seed or olive) oils. The effect of the variation of the carbonisation temperature (500, 650 or 800 °C) and PAN/PMMA mass ratio (1:0, 6:1 or 2:1) on the oil recovery efficiency and oil-from-water separation ability is investigated. Although the great number of sorbent and oil properties involved usually hinders finding general trends easily, the main factors controlling the oil recovery efficiency and oil from water separation ability of the investigated membranes are identified.

## 2. Materials and Methods

### 2.1. Preparation of the Self-Supporting Paper-Like Fibrous Membranes

Sigma Aldrich supplied *N*,*N*-dimethylformamide (DMF, HCON(CH_3_)_2_, anhydrous: 99.8%, CAS No. 68-12-2), polyacrylonitrile ((C_3_H_3_N)_n_, purity: 99.9%, average molecular weight: 150,000 g mol^−1^, CAS No. 25014-41-9) and polymethylmethacrylate ([CH_2_C(CH_3_)(CO_2_CH_3_)]_n_, average molecular weight: 350,000 g mol^−1^, CAS No. 9011-14-7). All materials were used without further purification.

The sorbent membranes (Figure 1) were produced following the procedure illustrated in detail in a previous work [49]. As sketched in Appendix A, it consisted of: (1) the preparation of the spinnable solution; (2) electrospinning; (3) the oxidative treatment of the as-spun membrane (stabilisation); and (4) subsequent annealing in inert atmosphere (carbonisation). Table 1 reports the composition of the polymer solution and the stabilisation and carbonisation conditions.

The spinnable solution was prepared by dissolving polymer(s) in DMF. Electrospinning was carried out in air (relative humidity: 40%) at a temperature of 20 ± 1 °C by means of a CH-01 Electro-spinner 2.0 (Linari Engineering s.r.l., Florence, Italy). The PAN/DMF and PAN/PMMA/DMF solutions were fed at volumetric rates of 1.41 mL h^−1^ and 1.00 mL h^−1^, respectively. In the former case, the applied voltage was 17.2 kV; in the latter, it was 15.0 kV. The collection distance was kept constant at 11 cm. Stabilisation was carried in static air; carbonisation was operated at flow of 100 cc min^−1^ N_2_. Further details can be found in the Appendix A.

### 2.2. Characterisation of the Produced Membranes

After thermal processing, the produced membranes were analysed using scanning electron microscopy (SEM), micro-Raman spectroscopy (MRS), X-ray diffraction (XRD) and Brunauer–Emmmett–Teller (BET) analysis. Their texture and morphology was investigated by SEM. Lower magnification SEM images were acquired using a Phenom Pro-X (Deben, Suffolk, UK) scanning electron microscope, endowed with Fibermetric software for the automated measurements of the NF diameters. A FEI S-FEG XL30 microscope (Philips, Eindhoven, The Netherlands) was used to acquire higher magnification images. The carbonisation of the membranes was ascertained using MRS. Raman scattering excited by a solid-state laser operating at 2.33 eV (532 nm) was measured by using a NTEGRA–Spectra SPM spectrometer (NT-MDT LLC, Moscow, Russia), equipped with MS3504i 350 mm monochromator and ANDOR Idus CCD (Oxford Instruments, Belfast, UK). The scattered light from the sample was collected by a Mitutoyo 0.75 numerical aperture 100× objective. The use of a very low laser power (250 μW at the sample surface) prevented local heating of the samples and annealing effects. A D2 Phaser diffractometer (Bruker, Karlsruhe, Germany), equipped with a Ni β-filtered Cu-K_α_ radiation source (λ = 0.1541 nm), was used to record XRD patterns. The specific surface area was determined from N_2_ adsorption/desorption isotherms at 77 K using Tristar II 3020 apparatus (Micromeritics, Norcross, GA, USA) and the instrumental software (version 1.03). Porosity distribution was determined by applying BET and Barrett–Joyner–Halenda (BJH) analyses. Before measurements, samples were pretreated (150 °C, 4 h, N_2_) to remove adsorbed foreign species.

### 2.3. Contact Angle Measurements

To investigate the hydrophilicity/hydrophobicity of the membranes, a water drop was released on the sorbent surface and the contact angle was measured from images caught at the optical microscope using the experimental setup purposely assembled (Appendix A).

### 2.4. Liquid Adsorption Tests

The liquid adsorption capacity of the produced sorbents was evaluated via two tests: (1) immersion tests (in a single liquid, oil or water); and (2) spill tests (to evaluate the oil recovery efficiency from an oil/water mixture and the selectivity towards oil adsorption). For this purpose, distilled water (control liquid), vegetal (olive and sunflower seed) and mineral (motor) oils were investigated. Their main specifics are reported in Appendix A.

#### 2.4.1. Immersion Tests

Immersion tests were carried out following the procedure illustrated in Appendix A. Sorbent, preliminarily weighed, was immersed in a glass beaker containing liquid (oil or water). After about 5 min of sorption, the sorbent was drained for 3 min, to remove the residual liquid droplet (although less than 1 min was enough to ensure the removal of excess of droplets). As usual [40], the sorption capacity, that is the liquid recovery efficiency, *R*_L_ (g g^−1^), of the membrane was determined via Equation (1):(1)$$R_{L}=\frac{M_{L}}{M_{0}}$$
where *M*_L_ (g) indicates the mass of adsorbed liquid, obtained by difference, *M* − *M*_0_, between the masses of the liquid-saturated sorbent after drainage (*M*) and of the dry sorbent (*M*_0_). After the adsorption test, the oil was removed from the sorbent by washing in organic solvent (diethyl-ether).

#### 2.4.2. Spill Tests

Appendix A illustrates the procedure of spill tests. About 5 mL of vegetal or mineral oil were added to 20 mL of water (to obtain a thin oil layer over the water), in order to simulate an accidental oil spill. After weighing (*M*_0_), sorbent was placed in the mixture of oil–water for 5 min and subsequently drained for 3 min to remove the residual liquid droplet (Appendix A), as in the immersion tests. After weighing (*M*), the drained sorbent was placed in a thermostatic chamber at 60 °C for the slow removal of water. After weight stabilisation, it was weighed again (*M*_1_). As in Wu et al. [50], the oil recovery efficiency, *R*_O_, was calculated using Equation (1) with the mass of recovered oil, evaluated as *M*_1_ − *M*_0_. The mass of water recovered was estimated as the difference between the masses of the sorbent before and after drying (*M* − *M*_1_); the corresponding recovery efficiency *R_W_*, was calculated via Equation (1). Finally, following Piperopoulos et al. [42], the oil-to-water selectivity of the sorbent (*S*_O/W_) was evaluated by normalising the oil-to water-recovery efficiency, namely:(2)$$S_{O/W}=\frac{R_{O}}{R_{W}}$$

## 3. Results and Discussion

### 3.1. Effect of the Thermal Treatments

Stabilisation in-air of as-spun membranes converts PAN to an infusible stable ladder polymer [51,52]. This happens through the cyclisation of nitrile groups (C≡N) and cross-linking of the chain molecules in the form of –C=N–C=N–, which limits volatilisation and maximises the carbon yield, and at the same time prevents melting during the subsequent step at higher temperature. The evolution of water vapour accompanies these chemical reactions. In the present case, regardless of the composition of the spinnable solution, stabilisation produced an average mass loss of 11.4 ± 1.7 wt%.

During carbonisation, the dehydrogenation and condensation of cyclic structures occur between 400 and 500 °C [53]. At carbonisation temperature (*T*_C_) ≥ 600 °C, graphite-like domains, with preferential alignment of the (002)-planes parallel to the NF axis, form on the lateral surfaces [52], while non-carbonised components are selectively removed from the NFs [51,53] under the form of volatile compounds, such as HCN, N_2_, NH_3_, CO and CO_2_, with consequent changes in the surface chemistry (including polarity) and morphology of the NFs (pores and beads). The carbonisation of the PAN-based membranes was accompanied by a mass loss roughly proportional to *T*_C_ (Appendix A), while during the carbonisation of the PAN/PMMA-derived membranes at 650 °C, the mass loss was roughly proportional to the relative amount of PMMA (Appendix A), which completely degrades below this temperature via homolytic scission of the chain, side group scission and depolymerisation [54].

### 3.2. Physico-Chemical Properties of the Sorbent Membranes

#### 3.2.1. Morphology and Texture of the NFs

The morphology of the thermally processed membranes was assessed via SEM analysis. Regardless of their preparation conditions, the membranes exhibited a non-woven fibrous morphology and were characterised by a great spatial homogeneity (Figure 1f). In the PAN-based membranes (Figure 2a,b,d), NFs were smooth; on the contrary, in the PMMA/PAN-derived membranes (Figure 2c,e), the surface looked rough, particularly for lower PMMA content (Figure 2e). Moreover, the coalescence of some NFs generated macropores (Figure 2e).

Figure 2f−i shows the distribution of the NF diameters, automatically calculated by the image analysis software. As expected [52], with increasing *T*_C_ (from 500 to 800 °C), the diameter of the PAN-based NFs slightly shrunk (the distribution centre value decreased from 360 to 310 nm). The blend-derived NFs, carbonised at the same *T*_C_ (650 °C), had larger diameters, in agreement with the literature reports [55]. The diameter increase was due to the generation of a relatively large volume of volatile products from PMMA degradation/gasification during the PAN carbonisation, which counteracted the NF contraction [43].

#### 3.2.2. Porosity

Figure 3 shows the N_2_ adsorption/desorption isotherms of the NFs and the pore size distributions. The isotherms of membrane P1:0-500 (Figure 3a) were different from those of the membranes carbonised at higher *T*_C_; they were featured by a steep slope at higher relative pressures, as typical of the presence of mesopores (2−50 nm) and macropores (>50 nm) [56]. 

Actually, the membrane carbonised at 500 °C had the smallest fraction of micropores (<2 nm, see *f*_micro_ in Table 2). At higher *T*_C_, the selective removal of non-carbonised components from the NFs as volatile species [51,53] led to a remarkable increase in *f*_micro_. In addition, in the membranes P1:0-650 (Figure 3b) and P1:0-800 (Figure 3c), the hysteresis occurred until relatively low relative pressures pointed at the formation of ultramicropores (<0.7 nm, non-quantifiable by nitrogen, [57]). The isotherms of the blend-derived membranes were different from each other (Figure 3d,e). Membrane P6:1-650 had greatly smaller surface area (*S*_T_) and micropore volume (*V*_micro_) compared to the blend-derived membrane carbonised at the same *T*_C_ (Table 2), hinting at a hindrance to the formation of open porosity; with increasing PMMA content of the blend, *S*_T_ and *V*_micro_ increased, and the occurrence of hysteresis (Figure 3e) pointed at the formation of ultramicropores as in membranes P1:0-650 and P1:0-800.

#### 3.2.3. Amorphous Nature of the NFs

Figure 4 displays the micro-Raman spectra of the membranes. The detection of the broad D- and G-bands (at 1346 and 1580 cm^−1^, respectively), typical of highly disordered graphitic nanocarbons [59], confirmed the polymer carbonisation, as well as the amorphous nature of the formed NFs. Graphitisation takes place only over 2500 °C [52]; at lower *T*_C_, graphite-like domains are formed [53], but the entanglement of PAN molecules in the electrospun NFs limits the development of crystals during carbonisation, resulting in low graphitic mole fraction [60]. The selective removal of the non-carbonised components from the NFs as volatile compounds [51,53] causes the composition of the polymer-based NFs to change [51,52].

As is known [49,58,61], the frequency position of the G-band (ω_G_) is sensitive to the presence of heteroatoms in the graphite-like lattice, while the D/G intensity ratio (*I*_D_/*I*_G_) is affected by the average size of the graphite-like domains (*L*_C_). In nitrogen-rich carbon NFs, the G-band downshifts [49], while in oxygen-functionalised carbons, it upshifts as an effect of the electron transfer from the π-states to the oxygen atoms [62]. Hence, the spectra were fitted in order to monitor the changes produced by the variation of *T*_C_ and PMMA amount. Following the most recent trends on the analysis of Raman spectra of amorphous nanocarbons [63], Gaussian–Lorentzian lineshapes were used to reproduce the spectra. The main results obtained are reported in Appendix A. The increase in *T*_C_ from 500 to 800 °C caused only a very slight increase in *L*_C_ (from 2.75 to 2.89 nm). This finding, in agreement with the literature reports [51,52] and the indications emerging from the XRD analysis (Appendix A), reflected the small change of the graphitic mole fraction taking place in this *T*_C_ range [60]. ω_G_ changed to a larger extent (from 1578 to 1588 cm^−1^), reflecting the decrease (increase) in nitrogen (oxygen) content in the NFs [64]. The increase in PMMA amount was also accompanied by a small *L*_C_ increase (from 2.84 to 3.01 nm); the upshift of the G-band (from 1581 to 1591 cm^−1^) signalled the increase in the oxygen content of the NFs [62].

### 3.3. Sorption Performance of the Membranes

#### 3.3.1. Water Immersion Tests

Figure 5 displays the liquid (water or oil) recovery efficiencies measured in immersion tests, while the numerical values are reported in Appendix A. Among the PAN-based membranes, the highest water recovery efficiency (*R*_W_) pertained to P1:0-500 that exhibited a hydrophilic nature. Water drop was immediately adsorbed on this membrane (Appendix A). The increase in *T*_C_ caused the membranes to become hydrophobic (Appendix A). In the membrane P1:0-650, the measured water contact angle was 135° (Appendix A), in agreement with the literature [44,65]. A significant decrease in *R*_W_ (from 78.9 g g^−1^ in membrane P1:0-500 to 11.5 g g^−1^ in P1:0-800) accompanied these changes (Figure 5a). At *T*_C_ = 650 °C (Figure 5b), *R*_W_ varied non-monotonically with the PMMA content of the blend, increasing in the order P6:1-650 (2.5 g g^−1^) < P1:0-650 (18.0 g g^−1^) < P2:1-650 (87.6 g g^−1^). 

As is well known [49,65,66,67], the presence of (nitrogen- and/or oxygen-containing) functional species on the material surface can noticeably increase its wettability in the presence of polar liquids. Actually, *R*_W_ was found to increase as the G-band downshifted or upshifted, signalling the changes in nitrogen and oxygen contents in the graphite-like carbon network (Figure 6).

#### 3.3.2. Oil Immersion Tests

Figure 5 displays the oil recovery efficiencies (*R*_O_) measured in immersion tests, while the numerical values of *R*_O_ are reported in Appendix A. Two different kinds of oils were tested: motor oil (MO) and vegetable oils (VOs), namely olive oil (OO) and sunflower seed oil (SFO). Their main specifics are reported in Appendix A. MO is typically composed (up to 73–80% *w*/*w*) of aliphatic hydrocarbons (primarily alkanes and cycloalkanes with one to six rings), 11–15% mono-aromatic hydrocarbons, 2–5% di-aromatic hydrocarbons and 4–8% poly-aromatic hydrocarbons [68]. Triglycerides are the main components (95–98%) of VOs [69]. Triglycerides are composed of fatty acids [70], which, according to their saturation level, are classified into saturated, mono- and poly-unsaturated fatty acids. The fatty acid composition of triglycerides in vegetable oils noticeably varies, and this variability determines different oil physical and chemical properties. The fatty acid chain length may slightly influence the polarity of oils. Minor components (less than 5%) are also present in vegetable oils, such as glycerolipids, phospholipids, and non-glycerolipids, including sterols, tocopherols/tocotrienols, free fatty acids, vitamins, pigments, proteins and phenolic compounds [71,72]. Most of these minor components have a polar character [73,74].

Generally, since hydrophobic surfaces are oleophilic (i.e., these surfaces show a stronger affinity to oils compared to water), they are preferred for oil absorption [43]. Accordingly, among the PAN-based membranes, P1:0-650 showed the highest *R*_O_ values for OO and MO (Figure 5a). 

The capacity of adsorbing MO shown by this membrane was comparable to the value reported by Liu et al. [29] for macroporous carbon NFs (67.8 g g^−1^ against 64.0 g g^−1^ for the latter). This capacity was greater than the value measured by Matripragada et al. [43] for carbon NFs with internal porous structures (44.9 g g^−1^). The blend-derived membranes exhibited *R*_O_ values (Figure 5b) always lower for the PAN-based membrane carbonised at the same temperature (4.4–42.8 g g^−1^ against 43.7–67.8 g g^−1^, respectively).

However, as is known [10,29,42], the chemical and physical properties of the oils also influence the adsorbing capacity of the membrane. Although VOs are classified as non-polar liquids, their polarity can be slightly influenced by the length of fatty acid chains and by the presence of minor components with polar character [73,74]. Generally, higher surface tension characterizes polar substances compared to their non-polar counterparts. 

A good correlation was found between *R*_O_ measured for the membrane P1:0-500, containing ~21 wt% nitrogen and being rich in polar surface-groups [49], and the oil surface tension (Appendix A): the higher surface tension of the oil, the higher *R*_O_.

For the remaining membranes, *R*_O_ was found to fairly correlate with the fraction of micropores (Figure 7), whose presence is known to be beneficial for capillary action to transport liquids [44,65]. In fact, for any oil, the higher *f*_micro_, the higher *R*_O_. The small differences observed were understood as the effect of the competition occurring between viscosity and surface tension [50]. On one hand, high viscosity can increase the sorption capacity by improving the adherence of oil onto the sorbent surface; on the other hand, it can hinder sorption, inhibiting the oil penetration into the smaller pores and inner parts of the sorbent. Low values of surface tension favour the oil penetration/trapping into the sorbent. The viscosity of the considered oils increases in the order SFO < OO << MO, while surface tension varies in the order MO < OO < SFO (Appendix A). The lower viscosity of VOs allowed for their adsorption into the parts of the sorbent, where the oil droplets have to deform to larger extents to penetrate, resulting in higher *R*_O_ compared to MO in membranes with smaller *f*_micro_. Conversely, the lower surface tension and higher viscosity of MO favoured its penetration and trapping into the sorbent due to the improved adherence to its surface, which accounted for the higher *R*_O_ compared to VOs measured for this oil in the presence of higher *f*_micro_.

#### 3.3.3. Spill Tests and Oil/Water Selectivity

Appendix A reports the numerical values of recovery efficiencies measured in the spill tests. Among the tested membranes, P1:0-650 showed the highest *R*_O_ values for any type of oil (reaching 100.3 g g^−1^ for OO, Figure 8a). In the case of MO, its recovery efficiency was five-fold and over the value reported by Alassod et al. [16] for the lower cost polypropylene melt blown non-woven membranes (68.7 g g^−1^ against 13.3 g g^−1^). Compared to the latter, the blend-derived membranes were always less efficient in recovering oil (Figure 8b).

As shown in Figure 8, in the spill tests, water was also recovered together with oil. The proportionality between the recovered water and recovered oil (Appendix A) strongly pointed at the encapsulation of water nanodroplets by oil, which could be an effect of the (mixing and) interaction between the two liquids during the immersion of the membrane in the bi-phasic mixture [75] (Appendix A).

By following Piperopoulos et al. [42], the oil/water selectivity (*S*_O/W_) of all membranes was evaluated via Equation (2). The results obtained are displayed in Figure 9 and reported in Appendix A. The water uptake accompanying the adsorption of VOs proceeded at a lower rate compared to MO (Appendix A), which resulted in higher *S*_O/W_.

Due to the many factors involved, finding general trends for *S*_O/W_ is a very difficult task. In the case of membrane P1:0-500, the strength of capillary force, as monitored by the ratio of surface tension to mass density of the oil (Appendix A), played a key role in the competition between the two liquids for the adsorption: the higher the capillary force, the lower *S*_O/W_. For the other membranes (Figure 10), *S*_O/W_ appeared to be mainly controlled by the average pore size (*d*_P_). However, the details of the *S*_O/W_ dependence on *d*_P_ were different: a monotonical increase with *d*_P_ was observed for MO, while the curves of VOs peaked around an optimal *d*_P_ value. This finding indicated that also, the oil properties matter, as is obvious. In membranes P1:0-650 and P1:0-800, featured by smaller *d*_P_ and higher *f*_micro_ (Table 2), *S*_O/W_ was found to decline with increasing oil viscosity (Appendix A). The higher oil-from-water separation ability pertained to membranes P6:1-650 and P2:1-650, featured by larger *d*_P_. Overall, among the investigated membranes, P2:1-650 looked to be the best performing in terms of both oil sorption capacity (73.5, 54.8 and 12.5 g g^−1^ for olive, sunflower seed and motor oil, respectively) and oil-from-water separation ability (74, 69 and 16 for olive, sunflower seed and motor oil, respectively).

## 4. Conclusions

Self-supporting paper-like PAN/PMMA-derived fibrous membranes were prepared via electrospinning. The as-carbonised membranes were evaluated as oil sorbents without carrying out any post-synthesis treatment with harsh chemicals. The effect of the variation of the carbonisation temperature (500–800 °C) and PAN/PMMA mass ratio (1:0–2:1) on the pore size distributions, microstructural properties, oil recovery efficiency and oil-from-water separation ability of the membranes was investigated. In spite of the great number of (sorbent and oil) properties involved, for all the membranes except those carbonised at the lowest temperature, the fraction of micropores was found to chiefly control the oil recovery efficiency in immersion tests, whereas the oil-from-water separation ability in spill tests was mainly dependent on the average pore size.

Thanks to the scalability of the electrospinning and the availability of commercial machineries able to produce fibre sheets up to 300–600 m^2^ large (depending on the sheet thickness), this laboratory-scale study provides a new perspective towards the sustainable manufacture of fibre sorbents on an industrial scale.

## Figures and Tables

**Figure 1 membranes-11-00515-f001:**
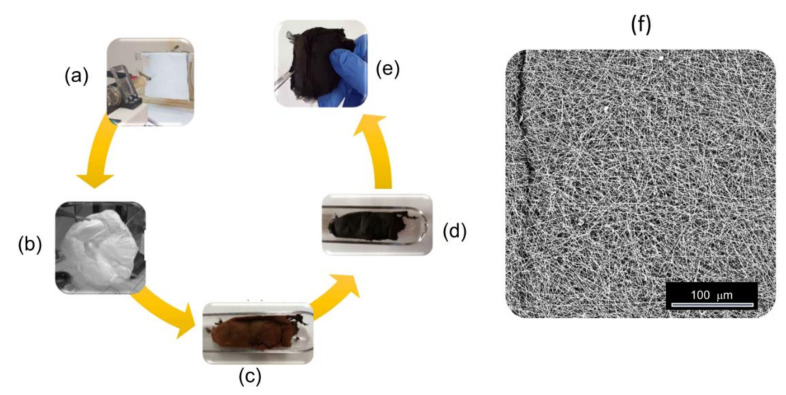
Process for the production of the sorbent membranes: (**a**) as-spun membrane after (**b**) peeling from the collector, (**c**) stabilisation and (**d**) carbonisation, (**e**) as-obtained self-supporting paper-like membrane and (**f**) its fibrous microstructure as evidenced by SEM.

**Figure 2 membranes-11-00515-f002:**
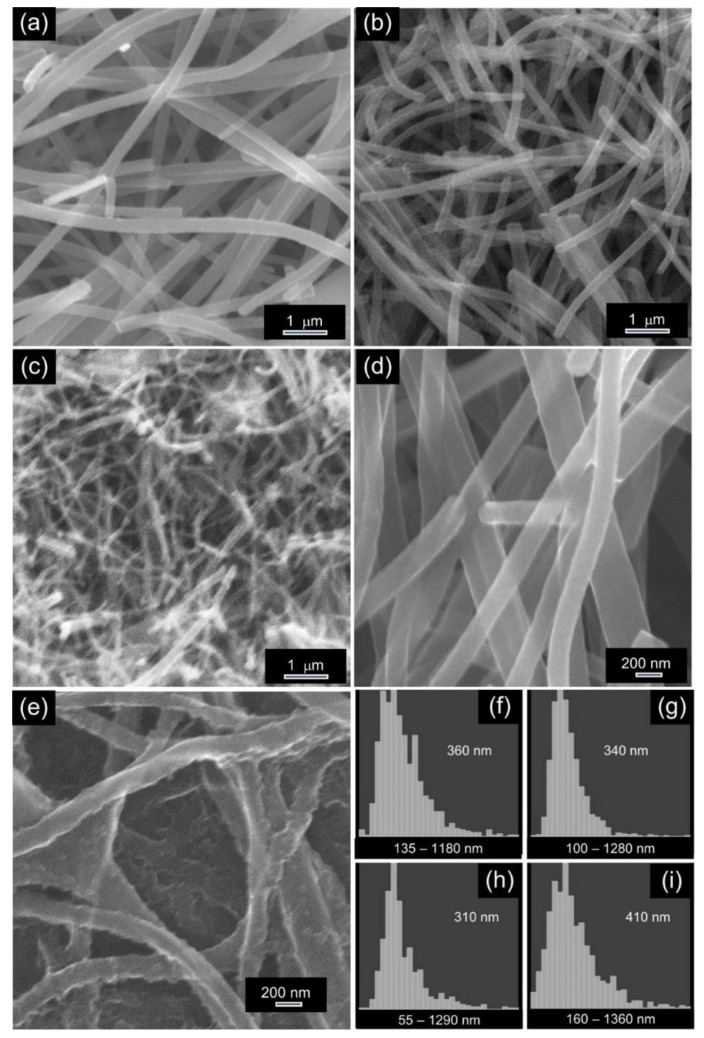
(**a**–**e**) Morphology of the fibrous membranes and (**f**–**i**) distributions of the fibre diameters. The shown SEM images refer to membranes (**a**) P1:0-500, (**b**,**d**) P1:0-800, (**c**) P2:1-650 and (**e**) P6:1-650. Histograms refer to samples (**f**) P1:0-500, (**g**) P1:0-650, (**h**) P1:0-800 and (**i**) P2:1-650 (the diameter distribution centre value is reported).

**Figure 3 membranes-11-00515-f003:**
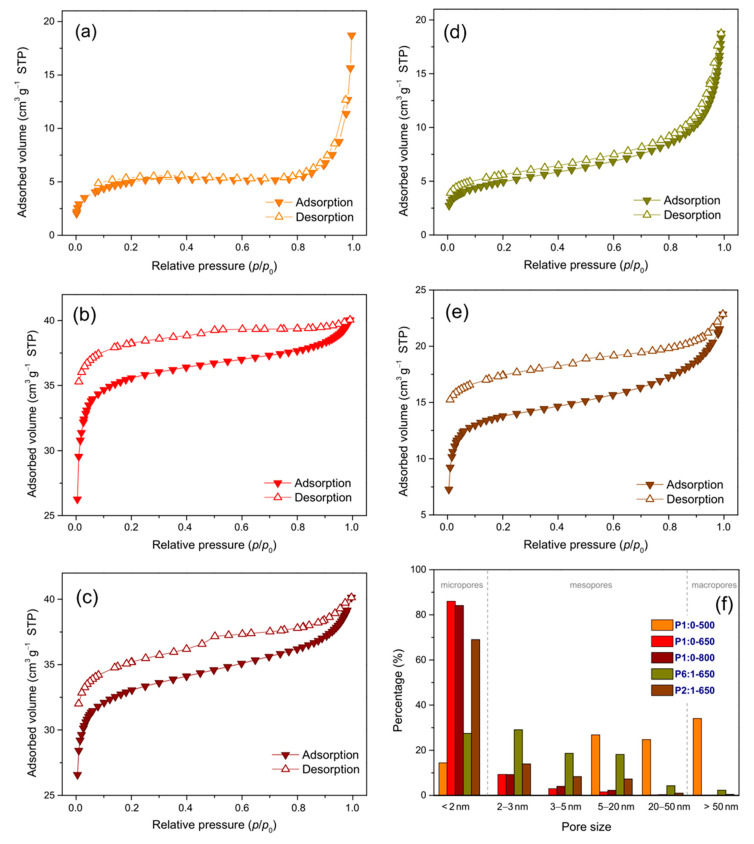
Nitrogen sorption isotherms of membranes (**a**) P1:0-500, (**b**) P1:0-650, (**c**) P1:0-800, (**d**) P6:1-650 and (**e**) P2:1-650, and (**f**) pore size distributions.

**Figure 4 membranes-11-00515-f004:**
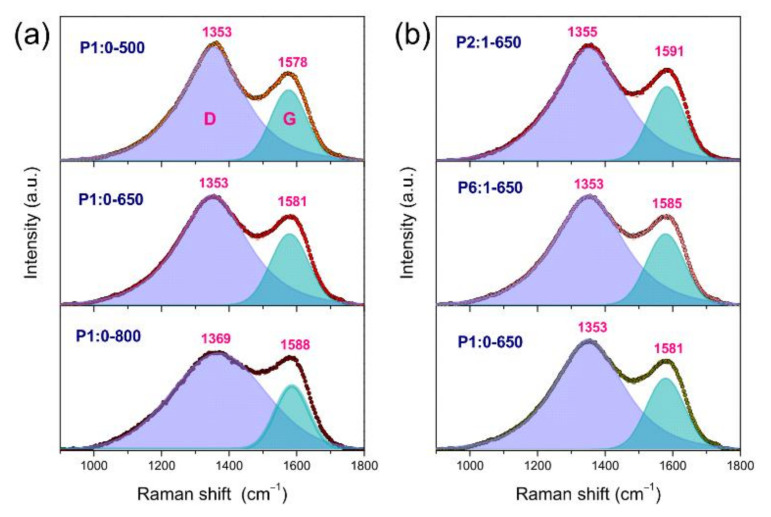
Micro-Raman spectra of (**a**) PAN-based membranes carbonised at different temperatures and (**b**) PAN/PMMA-derived membranes with different amount of PMMA carbonised at 650 °C.

**Figure 5 membranes-11-00515-f005:**
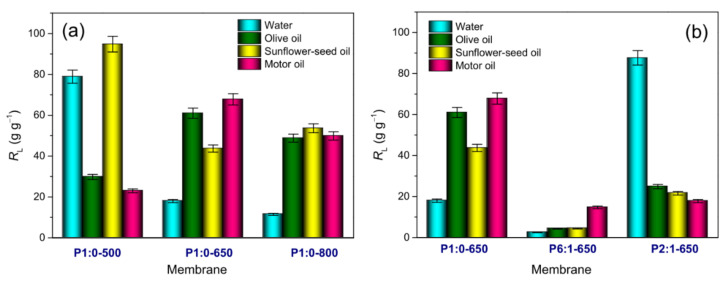
Liquid recovery efficiencies (*R*_L_) measured in immersion tests: effect of (**a**) the carbonisation temperature of PAN-derived membranes and (**b**) PAN:PMMA mass ratio of PAN/PMMA-derived membranes carbonised at 650 °C. For an easier comparison, in the two plots, the same vertical scale is used.

**Figure 6 membranes-11-00515-f006:**
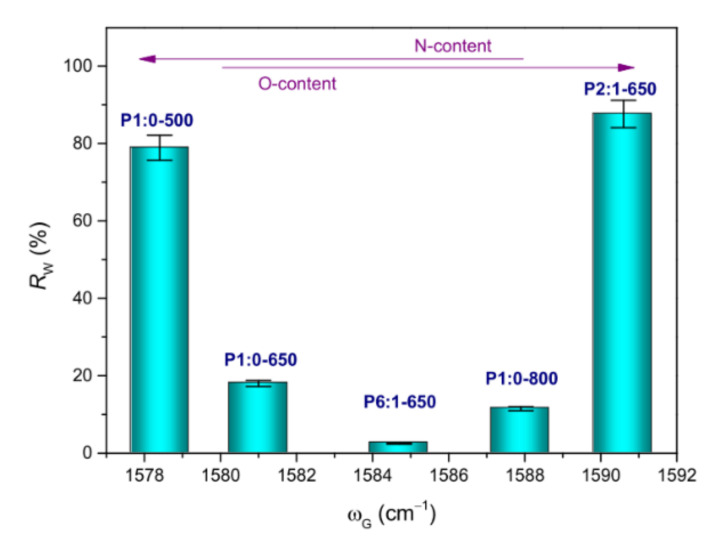
Water recovery efficiency (*R*_W_) as a function of the G-band frequency position (ω_G_).

**Figure 7 membranes-11-00515-f007:**
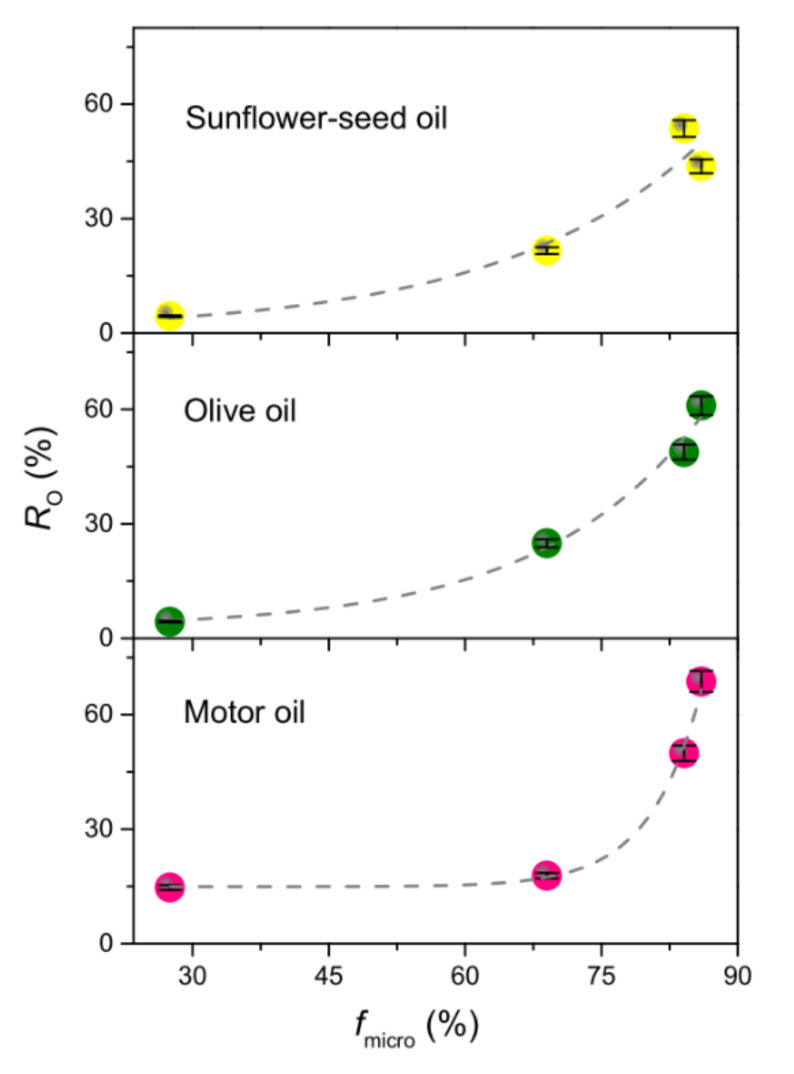
Oil recovery efficiency (*R*_O_) as a function of the micropore fraction (*f*_micro_).

**Figure 8 membranes-11-00515-f008:**
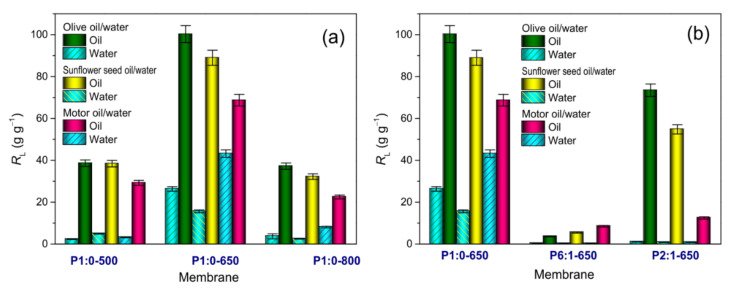
Liquid recovery efficiencies (*R*_L_) measured in spill tests: effect of (**a**) the carbonisation temperature of PAN-derived membranes, and (**b**) PAN:PMMA mass ratio of blend-derived membranes carbonised at 650 °C. For an easier comparison, in the two plots, the same vertical scale is used.

**Figure 9 membranes-11-00515-f009:**
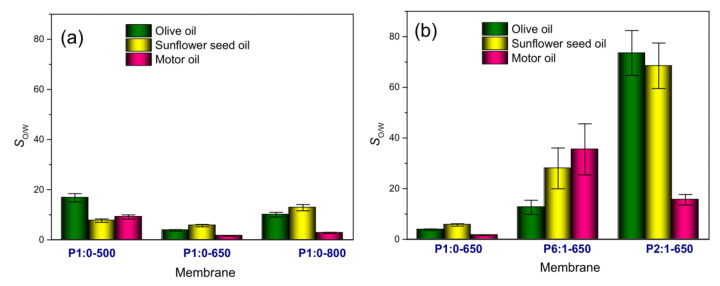
Oil/water selectivity (*S*_O/W_): effect of (**a**) the carbonisation temperature of PAN-derived membranes, and (**b**) PAN:PMMA mass ratio of blend-derived membranes carbonised at 650 °C. For an easier comparison, in the two plots, the same vertical scale is used.

**Figure 10 membranes-11-00515-f010:**
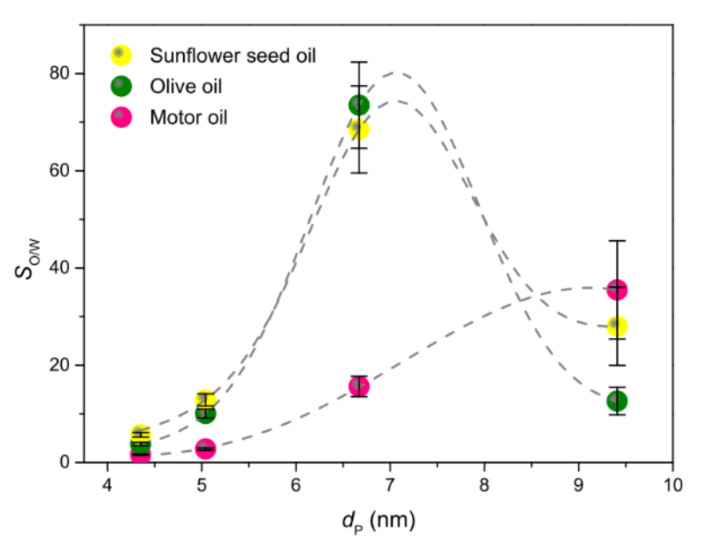
Oil/water selectivity (*S*_O/W_) as function of the average pore size (*d*_P_). Lines are drawn to guide the eye.

**Table 1 membranes-11-00515-t001:** Polymer(s) in the spinnable solution and post-electrospinning treatment conditions (*T*_S_ and *T*_C_ denote stabilisation and carbonisation temperatures, respectively; *t*_S_ and *t*_C_ indicate the duration of the processes). Temperature increasing rate was 5 °C min^−1^. Both the treatments were followed by uncontrolled cooling down to room temperature (RT).

Sample Code	Polymer(s) in the Spinnable Solution		Post-Electrospinning Heat Treatments	
	PAN:PMMA	PAN + PMMA (wt%)	*T*_S_ (°C)/*t*_S_ (h)	*T*_C_ (°C)/*t*_C_ (h)
P1:0-500	1:0	6.5	280/3	500/3
P1:0-650				650/3
P1:0-800				800/3
P6:1-650	6:1			650/3
P2:1-650	2:1			

**Table 2 membranes-11-00515-t002:** Specific surface area (S_T_), specific micropore surface area (S_micro_), specific micropore volume (V_micro_) and average pore size (d_P_) estimated by BET analysis. The fractions of micro-, meso- and macro-pores are also reported (f_micro_, f_meso_ and f_macro_, respectively).

Sample Code	*S*_T_ (m^2^ g^−1^)	*S*_micro_ (m^2^ g^−1^)	*V*_micro_ (mm^3^ g^−1^)	*f*_micro_ (%)	*f*_meso_ (%)	*f*_macro_ (%)	*d*_P_ (nm)
P1:0-500	20.3 ± 0.1	2.9	2.57	14.4	51.5	34.1	4.37
P1:0-650	139.7 ± 0.9	120.1	46.41	86.0	13.9	0.1	4.35
P1:0-800	128.8 ± 0.6	108.3	42.13	84.1	15.8	0.1	5.04
P6:1-650	17.5 ± 0.1	4.8	2.08	27.5	70.2	2.3	9.41
P2:1-650	53.3 ± 0.2	36.8	14.05	69.0	30.5	0.5	6.67

Note: in the presence of hysteresis, the adsorption isotherm that corresponds to more stable thermodynamic conditions [58] was utilised to evaluate the membrane porosity.

## Data Availability

The data that support the findings of this study are available from the corresponding authors upon reasonable request.

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
