# Peer review of "Evaluation of Electrospun Self-Supporting Paper-Like Fibrous Membranes as Oil Sorbents"

_membranes, 2021, doi:10.3390/membranes11070515_

Round 1

Reviewer 1 Report

  1. Abstract: Please add quantitative results and provide appropriate reasoning.
  2. What is the significance and outcome of this study?
  3. [Experimental section] The remarks information for chemicals and testing equipments in this work should be given, for example ( Model, Name of supplier, City, Country)
  4. [Experimental results and discussion ] all figures in the section should be rechecked seriously and refined. For example, scale bar and error bar is necessary.
  5. [Experimental results and discussion ] figure 2 ,  c and e images are not clear. Please add the clear images.
  6. Avoid the typo errors and unnecessary space in the written manuscript.

Author Response

Please see the letter attached

Reviewer 2 Report

Manuscript: Evaluation of electrospun self-supporting paper-like fibrous membranes as oil sorbents

Manuscript presents good research work related to water purification by using natural fibers. It is recommended for publication after following minor changes.

  • Abstract should contain some quantitative information also.
  • English must be improved.
  • Novelty of the work be established.
  • All the results reported be compared in a tabular form to establish the superiority of the work.
  • Authors must need to incorporate some recent references related to use of natural fibers for water purification in the introduction part of the manuscript to make it more interesting for the readers. For example;

 Environmental Science: Water Research & Technology 6 (11), 3080-3090

Advanced Sustainable Systems, 1900114

RSC advances 9 (69), 40565-40576

            ACS Sustainable Chemistry & Engineering 6 (3), 3279-3290

         Industrial & Engineering Chemistry Research 56 (46), 13885-13893

  • Authors must need to incorporate future prospective of the presented work in the conclusion part of the manuscript.

Author Response

Please see the letter attached
